# A gendered content analysis of the World Health Organization's COVID-19 guidance and policies

Emily Tomsick[1], Julia Smith[2]*, Clare Wenham[3]

**1** School of Public Policy, University of Maryland, College Park, Maryland, United States of America,
**2** Faculty of Health Sciences, Simon Fraser University, Burnaby, British Columbia, Canada, **3** Department of Health Policy, London School of Economics and Political Science, London, United Kingdom

* jhs6@sfu.ca

**Data Availability Statement:** Data is available in a supplementary file to this submission - and are freely available on WHO website (https://www.who.int/emergencies/diseases/novel-coronavirus-2019/who-response-in-countries).

## Abstract

As with previous global public health emergencies, the COVID-19 pandemic has had distinct and disproportionate impacts on women and their health and livelihoods. As the leader in global public health, it is incumbent upon the World Health Organization (WHO) to ensure gender is prioritized in pandemic response. We conducted a policy analysis of 338 WHO COVID-19 documents and found that only 20% explicitly discuss gender and over half do not mention women, gender, or sex at all. Considering the well documented gendered effects of pandemics and the WHO's commitment to gender mainstreaming, this paper: 1) asks to what degree and how the WHO incorporates a gender inclusive approach; 2) maps where and how gender considerations are included; and 3) analyses what this suggests about WHO's commitment to gender mainstreaming within its COVID-19 response and beyond. We demonstrate that WHO should increase its gender mainstreaming efforts and incorporate gender considerations related to health emergencies more often and in more policy areas.

## Introduction

On 30th January 2020, the World Health Organization (WHO) declared the novel coronavirus to be a Public Health Emergency of International Concern (PHEIC) [1]. Alongside the PHEIC declaration, the organization produced initial temporary recommendations as to what governments and the broader global health community should be doing to prevent, detect and respond to the risk of the global circulation of the pathogen. As with previous epidemics, these focused on surveillance, surge testing, contact tracing, access to clinical supplies and PPE and support for healthcare workers. As part of the advice and technical documentation provided to states by the WHO as global epidemic coordinator, tasked with "directing and coordinating authority on international health work", the international organization subsequently published 338 documents in 2020, providing robust evidence and best practice guidance for member states on how to respond to COVID-19. These have varied in scale and content but are rooted in best public health guidance and evidence-based approaches to combatting the pathogen, as

**Funding:** JS and CW work was supported by the Canadian Institutes of Health Research (cihr-irsc.gc.ca) under grant OV7-170639. The funders had no role in study design, data collection and analysis, decision to publish, or preparation of the manuscript.

**Competing interests:** The authors have declared that no competing interests exist.

per their global public health mandate. For many governments around the world, these publications have provided the blueprint for responding to the pandemic.

The gendered effects of COVID-19, and the preceding epidemics, are by now well established. The lack of data disaggregated by sex has long been an issue in public health emergency response, recent examples include H1N1 in 2009 and Ebola in 2014–2015 [2–4]. What's more, globally, women comprise 70 percent of the healthcare workforce [5, 6] and are more likely to be in prolonged contact with patients [7], putting them at higher risk for infection and mortality, as well as the associated mental health burden of working on the frontline. In addition to being overrepresented in the healthcare workforce, women also are disproportionately represented in the care economy [8, 9]. The pandemic has caused an increase in domestic caregiving responsibilities to reduce the burden on overwhelmed healthcare systems and reduce disease transmission by closing schools, the effects of which have fallen on women [10]. Women are also more likely than men to work in industries that are negatively impacted by pandemic containment measures, including hospitality, food service, retail, and education [11, 12]. The disproportionate sector-specific effects on women, combined with the increase in caregiving responsibilities, have led to a situation in which women are more vulnerable to becoming unemployed or being forced to leave the workforce entirely than men. Access to sexual and reproductive healthcare (SRH), especially obstetric care, is disproportionately limited during public health emergencies, if such services are deemed to be "non-essential" [6, 7, 13–15]. Pandemics further exacerbate existing gender inequalities and vulnerabilities, increasing the risk of abuse, gender-based violence (GBV) and exploitation [5, 7, 16, 17]. Early evidence suggests that the improvements in visibility and positioning of gender being observed in the current pandemic response are mainly performative, narrowly focused on women's presence in leadership, and have not translated into policy solutions [18]. Whilst the global health community have recognized gender as a determinant for disproportionate suffering during the pandemic, minimal mitigations have been implemented to try and reduce the unequal burden placed on women by disease control interventions [19]. Similar to Global Health 50/50's policy tracker, UNDP launched a COVID-19 Global Gender Response tracker in September 2020 to assess the gender sensitivity of policy measures implemented by governments around the world. Initial analysis included 2,500 measures across 206 countries; and found that only 12 percent of those countries had pandemic-related measures that addressed gender meaningfully and 42 percent didn't have any measures in these areas at all [18].

Though the WHO does not have the legal authority to implement national policy, the WHO plays a crucial role in establishing norms and priorities in global health security, and therefore, potentially, in advancing a gender-responsive approach to COVID-19 [4]. In his remarks on 15 May 2020, the Director-General of the WHO requested that governments ensure a gender-inclusive and non-discriminatory pandemic response, including collecting sex-disaggregated data, preventing and responding to issues of domestic violence, encouraging the availability and access to sexual and reproductive healthcare, protecting and supporting all health workers, and ensuring equitable access to COVID-19 testing and treatment [20]. Such themes were echoed in the organization's "Gender and COVID-19" advocacy brief. These build on WHO's previous commitment to "promote gender equality and to mainstream gender in all of the Organization's work" in its 13[th] General Programme of Work (2019–23), and its organizational gender strategy [21]. WHO adopts the UN definition of gender mainstreaming as, "Gender mainstreaming is the process of assessing the implications for women, men and gender diverse people of any planned action within a health system, including legislation, policies, programmes or service delivery, in all technical areas and at all levels." Considering the well documented gendered effects of pandemics and the WHO's commitment to gender mainstreaming, this paper: 1) asks to what degree and how the WHO incorporated a gender

inclusive approach; 2) maps where and how gender considerations are included; and 3) analyses what this suggests about WHO's commitment to gender mainstreaming within its COVID-19 response and beyond.

Terms such as gender sensitive, gender inclusive, gender-based and gender mainstreaming are often used without a clear definition [22]. Gender mainstreaming in particular has been defined in multiple ways, both allowing a flexibility that has enabled widespread adoption and a vagueness that can inhibit action and allow co-option. While the prominence of gender mainstreaming across global institutions and policy spheres over the last two decades since the Beijing Conference has been celebrated, the disconnect between policy adoption and lack of progress in addressing gender inequalities has raised questions about the concept's transformative potential [23]. Feminist scholars have argued gender mainstreaming has not delivered on its promises, has failed in its primary objectives, and even impeded progress towards gender equality [24]. Yet, gender mainstreaming as an overarching policy approach has endured, maintaining space, however limited, to challenge inequities and propose alternatives. As noted above, the response to COVID-19 in particular has seen an unprecedented degree of attention focused on the gendered effects of pandemics, providing an opportunity to revisit the potential and limitations of gender mainstreaming.

Within global health, gender mainstreaming has often been narrowly defined, applied primarily to initiatives related to reproductive health, HIV/AIDS and GBV, and focused on the needs of women's health; as opposed to deconstructing unequal power relationships or engaging with the broader social, political and economic determinants of health that the feminist movement which proposed gender mainstreaming is concerned with [23]. Situated with global health's positivist biomedicalism, gender mainstreaming within global health institutions is often reduced to collecting sex disaggregated data or advocating for gender parity in decision-making [25]. One analysis of UN agencies working on global health describes progress on institutional gender mainstreaming as "modest" due to lack of clear road maps, inconsistent leadership support and entrenched power hierarchies [26]. Applied more as a technical than a political process, gender mainstreaming within global health has been critiqued for "delinking gender mainstreaming from social transformation and social justice agendas" [27].

Ravindran et al. note that in most WHO Regional Offices a single person is responsible for the gender, human rights and equity unit and gender allocations under the "Gender, Equity, Rights and Social Determinants" heading represent just 1.14% of the total WHO budget in 2017 [27]. The WHO's definition of gender mainstreaming (above) reflects a focus on process (i.e. "assessing implications" as opposed to rectifying inequities) and on "technical areas and levels", as opposed to political processes or power relationships. Recent analysis has further revealed tensions between the formal commitment to gender mainstreaming within WHO, and the failure to implement practices of gender mainstreaming throughout programmes [4]. Our analysis therefore takes this narrow definition of gender mainstreaming as a baseline, asking first if the WHO met its own standards of gender mainstreaming within the COVID-19 response, while keeping in mind calls for more transformative approaches.

## Methods

We conducted a gendered policy content analysis on all WHO documentation related to COVID-19, based on the READ methodology to systematically shift through all materials [28]. To ready our material, data was compiled through searches on WHO's website. We searched all "COVID_19" tagged publications on the website and in the WHO Publications Repository. Publications were limited to one year (2020) and included policy and advocacy briefs, health advisories, technical guidance, interim guidance, scientific briefs, country missions, survey

reports, research frameworks, case report forms (CRFs), and various surveillance tools and user guides. Joint publications co-authored by the WHO and other organizations were included. WHO regional office publications, country-specific situation reports and status updates, weekly epidemiolocal updates, journal articles, and corrigenda were excluded. The collection process resulted in a total of 338 documents to be included in the analysis.

Data was extracted from the documents in Microsoft Excel, whilst being analyzed for explicit mentions of "gender," "women," "girls," "female," "sex," and "pregnan*." This allowed us to see when the WHO expressly included mention of gender, women or sex in their policy documents and to understand how this was conceptualized by those launching the response. We were then able to further disaggregate these data sources between those which related to biological sex functions (e.g. sex disaggregated data, pregnancy, etc.) and those which consider broader gendered concerns. WHO defines gender as the characteristics of women, men, girls and boys that are socially constructed, including norms, behaviors and roles associated with being a woman, man, girl or boy, as well as relationships with each other. Sex, on the other hand, refers to the biological or physiological differences between men, women, and intersex persons. It is important to make these distinctions because any attempt to address the gendered dimensions of health must go beyond biological differences (sex) and include the socially and economically determined characteristics (gender) [29].

As gendered issues are often considered without explicit consideration or language on gendered determinants or effects, we also sought to analyze where such issues are considered, and how they are framed [30], through a second round of data extraction. Situating our analysis within the COVID-19 pandemic, we selected 10 thematic areas established in the literature on the pandemic to be gendered to establish the context for analysis. We then analyze how the WHO, as key actor with a normative leadership function in the COVID-19 response discussed these themes in the policy documents, if at all. The categories are pregnancy and maternal health, maintaining essential health services, ensuring access to sexual and reproductive healthcare (SRH), gender-based violence, legal status and displacement, education, healthcare workforce, unpaid care work, labour, and risk communication and community engagement (RCCE). One document can be counted in multiple categories, for example a document that explicitly discusses gender and references the healthcare workforce and gender-based violence would count toward all three of those categories. We recognize a limitation of our analysis is that it does not consider the process of policy development or implementation. Instead, the content analysis aims to depict where and how gender was included in WHO documents, as one form of evidence of if and how gender is mainstreamed within WHO's COVID-19 response.

With these documents, we then undertook a comprehensive documentary content analysis of the policy documents, seeking to understand if, where and how gender considerations where included. We asked what type of issues were identified as gendered and why, and how gender was framed in relationship to COVID-19 health challenges and solutions. We identified the most common topic areas including gender concerns, and patterns and differences in framing of gendered inequities, roles, and norms. We also noted where gender was not mentioned, particularly within those issue areas recognized as gendered in other literature and policy documents. We then sought to distill our findings amongst contemporary debates about the role of WHO in global health governance, and amid the discourse on gender and global health, recognizing that policy is not made in a vacuum, and that the context of the actor and contemporary environment and debates are important to the trajectory of policymaking [31].

The analysis was conducted by one team member with one reviewer who replicated the web searches. All team members selected a random sample of the same documents to analyze to ensure consistency of approach. The team had continual discussions throughout the process

to iteratively determine whether there were any borderline or gray areas, and whether or how to include them. All authors approved the final analysis framework. Where quotes are included below, these are exemplary, rather than exhaustive.

## Results

In May 2020, WHO advised countries "to incorporate a focus on gender into their COVID-19 responses in order to ensure that public health policies and measures to curb the epidemic take account of gender and how it interacts with other areas of inequality" [32]. Despite this advice, only 20% of the COVID-19 documents explicitly discuss gender and over half do not mention women, gender, or sex at all.

Table 1 shows the results of the analysis of explicit mentions of gender, biological sex only, and each gendered impact area. Out of the total 338 documents, sixty-four were found not to mention gender, sex, or any of the ten gendered impact areas in this analysis. The second column indicates the percentage of the total documents that fall into each category and the last column shows the percentage of documents within each category that also explicitly discuss gender. Fig 1 disaggregates where reflection was made as to gender, compared to sex, or neither within the documents analyzed.

There is an acknowledgment throughout the documents of the existence of underlying health and social inequities that lead to disproportionate impacts of public health crises. Some documents further explain that adherence to public health and safety measures and health-seeking behaviors are influenced by norms, gender ideologies, and power dynamics that exist in a given society. The advice found in two documents is to "conduct initial risk analysis and capacity assessment, including mapping of vulnerable populations, specific to the setting, to inform the operational plan, with a focus on reducing health and social inequalities that disproportionately affect women and girls" [33, 34].

The calls for gender-responsive policy that can be found in these documents address issues such as continuity of essential services, quarantine and vaccination plans, gender-based violence, and livelihoods. An example of such guidance can be found in the COVID-19 Strategy update from April 2020:

> National plans should also be developed for the prevention and mitigation of the social impacts of the crisis, including areas of the response that disproportionately affect women and girls. For example, many countries that have implemented restrictions on movement

**Table 1. Results of WHO COVID-19 policy analysis by gendered impact area.**

| Explicit mention / topic | No. of documents | Proportion of total documents (%) | Proportion that also discuss gender (%) |
|---|---|---|---|
| Gender | 67 | 19.82 | - |
| Biological sex | 89 | 26.33 | - |
| Essential health services | 56 | 16.57 | 58.93 |
| Sexual and reproductive health | 25 | 7.40 | 80.00 |
| Pregnancy/maternal health | 92 | 27.22 | 40.22 |
| Gender-based violence | 34 | 10.06 | 82.35 |
| Education | 37 | 10.95 | 54.05 |
| Healthcare workforce | 191 | 56.51 | 25.13 |
| Unpaid carework | 32 | 9.47 | 53.12 |
| Labour | 47 | 13.91 | 40.43 |
| Legal status | 46 | 13.61 | 41.30 |
| Risk communication and community engagement | 87 | 25.74 | 41.38 |

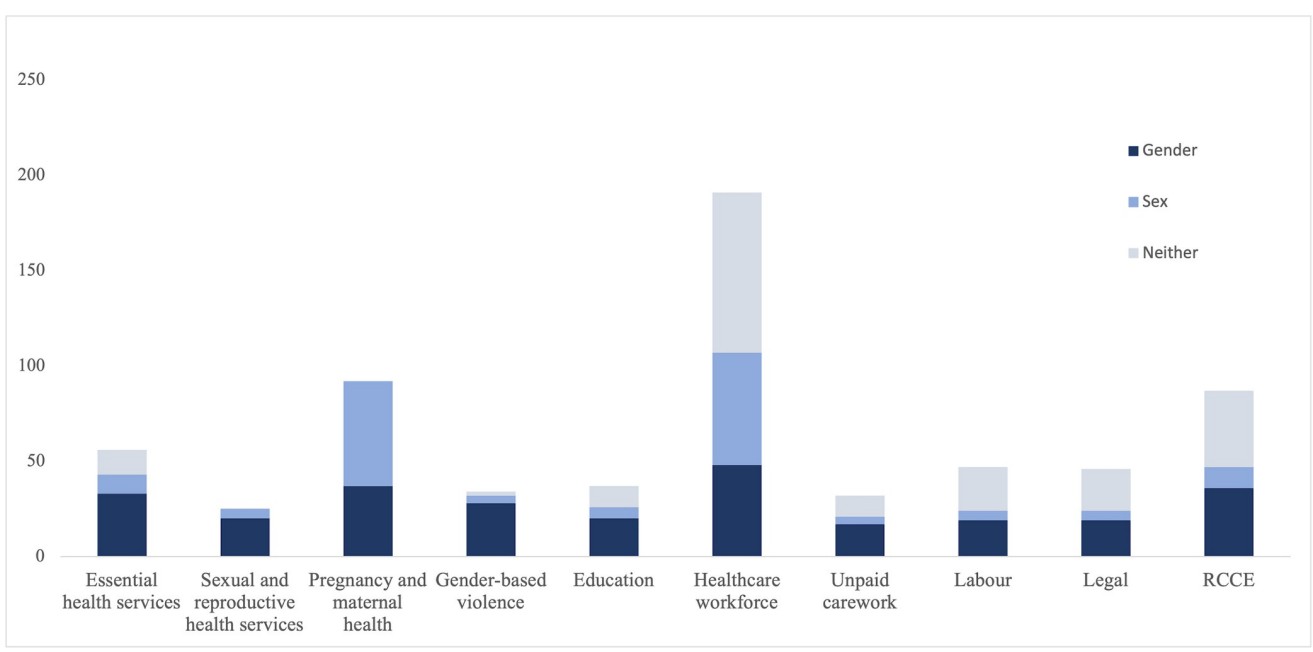

**Fig 1. Total documents by area & discussion of gender, biological sex only, or neither.**

outside of households have reported sharp increases in gender-based violence, primarily impacting women. Additionally, women are often most likely to be in insecure work and least likely to be covered by income-protection schemes, which are primarily designed for workers in formal employment. [35]

Another common theme is including women in the list of high-risk groups for whom the immediate and long-term effects of COVID-19 need to be taken into consideration. Though the dominant framing is to use the term "women and girls," with no mention of gender minorities or gender diverse people, a few documents use more inclusive language when referring to gendered risk or consideration. One example of this comes from the "WHO Concept for fair access and equitable allocation of COVID-19 health products" from September 2020:

All humans are of equal moral value and possess the same rights by virtue of being human. Consequently, vaccines should not be allocated on the basis of irrelevant characteristics over which humans have no control and which are arbitrary from the moral point of view. Such characteristics include race, colour, ethnic origin, place of origin, ancestry, citizenship, sex, gender identity and expression, sexual orientation, creed, family status, marital status, or ability to pay. [36]

Other documents use phrases like "sexual and gender minorities" to describe historically disenfranchised groups that could face disproportionate impacts of the pandemic due to existing stigma, discrimination, and barriers to healthcare and other essential services [37, 38].

The WHO emphasizes the importance of collecting sex-disaggregated data throughout its COVID-19 policy guidance. This emphasis is also present in the WHO's COVID-19 research guidance. The household transmission investigation protocol from March 2020 advises, "Any investigation of this nature should include reporting on the following information, stratified by age, sex, and relevant time and place characteristics" [39]. Approximately 21% of the

documents collected either call for the target audience to collect sex-disaggregated data or serve the purpose of collecting sex-disaggregated data itself, which is reflected in the (limited) data that is sex-disaggregated when returned to WHO. This includes the reporting of differential mortality rates, which have reportedly been higher for males compared to females, and the influence of sex and age on the severity of a COVID-19 infection. The language used in one document ties together the effects of biological sex and the gendered impacts of the pandemic to explain the differences and the intersections between them:

> Initial data demonstrate that men are more likely to suffer from severe COVID-19 than women. This is likely explained by a combination of factors including social, behavioural, genetic and hormonal factors, and differences in the biological pathways related to viral infection. Men have a higher frequency of underlying conditions, including cardiovascular disease, and are more likely than women to smoke. However, data from rapid gender assessment surveys suggest that women are particularly vulnerable to COVID-19. Women are more likely to be the caregivers and less likely to have access to health care and testing. In addition, health care workers are particularly at risk of contracting COVID-19, and women make up 70% of health care workers globally, and 80% of nurses in most regions. [40]

## Pregnant women and COVID-19

Pregnant women are frequently classified as a vulnerable population during COVID-19. The WHO typically refers to pregnant women clinically, and in many documents, the only time women are discussed at all is in terms of their pregnancy status. Examples include listing pregnancy as a pre-existing condition, as a category for data disaggregation and, in one document, as a group of people with mobility restrictions [41]. The WHO further published two documents specifically about breastfeeding and COVID-19. The prevailing recommendation given is that mothers with suspected or confirmed COVID-19 should be encouraged to initiate and continue breastfeeding, because the benefits far outweigh the potential risk of transmitting the virus through the mother's milk [42, 43]. The language used is consistently gendered as pregnant women (as opposed to pregnant people) and mothers (as opposed to parents).

The WHO also discusses pregnant women in terms of their inclusion in COVID-19 research. The prevailing narrative is that the decision to include or not include pregnant and lactating women should be made based on the relative risks and benefits, but that "Pregnant women and children should not be routinely excluded from research participation" [44]. In line with this recommendation, the WHO published a generic research protocol that is designed to study the potential impact of SARS-CoV-2on pregnant women [45]. The results of the study are meant to be used in advocating for women's access to reproductive health and maternal health services and inform the appropriate health and obstetric care for women who are or may be infected with the virus.

## Maintaining essential health services and access to sexual and reproductive health

When the WHO discusses maintaining essential health services during the COVID-19 pandemic, it primarily focuses on health facility capacity, service availability, risk reduction, and the dissemination of guidance and information to healthcare workers and home care providers. In line with its specific recommendations for health care facilities, the WHO published tools for assessing and monitoring capacity and service provision, "including essential medicines, diagnostics, equipment, safety measures and health care workforce capacity" [46].

The WHO's "Gender and COVID-19" advocacy brief discusses the potential differences between women and men that can impact their likelihood and ability to seek out essential healthcare services:

> Gender norms differently influence timely access to needed health services for both women (e.g. because of restricted autonomy in decision-making) and for men (e.g. because of rigid notions of masculinity that may delay timely care-seeking). Lockdowns and physical distancing measures are likely to exacerbate existing cultural restrictions on women's mobility, further limiting their access to and control over resources and their decision-making power in households. [32]

The WHO also conducted a pulse survey on the continuity of essential health services during the COVID-19 pandemic and found that family planning, contraception and antenatal care were disrupted in most countries [47], yet, fewer than half of the COVID-19 documents identified as discussing maintaining essential health services included express mention of SRH (23 out of 56).

Outside of documents that are already focused on women and gender, the framing of access to SRH tends to exclude the word "sexual" and instead focuses on reproductive health in terms of pregnancy and childbirth. For example, "countries should identify context-relevant essential health services that will be prioritized for continuation during the acute phase of the COVID-19 pandemic. High-priority categories include services related to reproductive health, including during pregnancy and childbirth" [48]. Access to contraceptives is typically discussed in terms of the disruptions to supply chains and as part of the overall limitations in access to healthcare and family planning services [49]. Only two documents explicitly discuss abortion [32, 50].

## Gender-based violence (GBV)

Approximately 10% of the documents analyzed discuss GBV, domestic violence, or intimate partner violence. Importantly this represents 42% of the documents that explicitly mention gender. The WHO acknowledges that public health and safety measures, such as mandatory stay at home orders, lead to increased risk of violence, particularly against women, children, and other marginalized populations. One document states, "Gender-based violence (GBV) increases during every type of emergency, including disease outbreaks. Care and support for GBV survivors may be disrupted, including safety, security and justice services" [38].

To address the issue of GBV during COVID-19, the WHO recommends that member states weigh the decisions to implement or alter public health measures against the negative impacts they have on society and individuals. It also recommends that governments "Work with domestic abuse prevention and civil society organisations, health care providers, local women and men, child protection groups, schools and youth organisations to promote programmes to reduce the risk of domestic violence, especially during periods of restricted movement" [51]. There are also multiple documents that lay out the potential role of the health sector and healthcare workers in identifying cases of GBV, providing frontline support, and facilitating access to support services [48, 50, 52].

## Legal status and displacement

Approximately 12% of the documents analyzed discuss legal status, displaced peoples, refugees, asylees, migration, ground crossings, points of entry, informal settlements, low capacity/ humanitarian settings, etc. The focus tends to be on risk management at points of entry,

including measures such as exit and entry screening, isolation for sick travelers, crowd control, physical distancing, mask use, and hand hygiene. The WHO recommends that countries "Develop and activate a COVID-19 emergency response plan where there are cross- border mass movements, such as displacement or migration. Response measures need to be tailored to the risk of COVID-19 spread, based on the epidemiological situation of the country/area of origin of the travellers" [53]. Some of the WHO COVID-19 documents are also specifically about refugees, migrant populations, and humanitarian settings, as indicated by their titles. These documents tend to acknowledge the disproportionate impact that women and girls in low-capacity settings will face such as violence, lack of social and financial protection, and decreased access to health care, especially to SRH services.

The WHO also includes gendered language and emphasizes the importance of not letting the COVID-19 situation further marginalize displaced peoples in advice to member states for managing camp settings:

> The right to COVID-19 preparedness, prevention and control for refugees and migrants should be exercised through non-discriminatory, child- and gender- sensitive comprehensive laws and national policies and practices. The health conditions experienced by refugees and migrants, including those with COVID-19 infections, should not be used as an excuse for imposing arbitrary restrictions, stigmatization, detention, deportation and other forms of discriminatory practices. [54]

The two joint publications on this topic are particularly gender-sensitive in their framing of the issues and their recommendations:

> Women and girls are likely to experience distinct challenges and risks associated with the COVID-19 outbreak, exacerbating already existing gender inequalities. Increased responsibility related to caregiving and within the household may limit women and girls' access to information and services. During an outbreak, where women have less power in decision making than men, their needs may largely be unmet and life-saving resources for reproductive and sexual health may be diverted to the emergency response. In addition, life-saving care and support to GBV survivors may be disrupted. Women and girls are also at heightened risk of intimate partner and other forms of domestic violence as a result of increased food insecurity and heightened tensions in the household. Negative economic impacts may increase the likelihood of survival sex, transactional sex and risk of sexual exploitation and abuse in the community and within projects, all of which greatly enhance exposure to the COVID-19 virus as well as sexually transmitted diseases. [55, 56]

This interim guidance publication also emphasizes the importance of considering gender norms and cultural aspects when formulating pandemic response plans. The other joint publication on COVID-19 readiness in refugee camps recommends that mitigation efforts, "be informed by a thorough gender and protection sensitive assessment of the impact on the wellbeing and satisfaction of basic needs of the men, women, boys and girls affected, and a plan for alternative provision of services and assistance to the individuals as well as the community must be prepared" [56].

## Education

While education is not a primary focus of the WHO, education and school closures appear in 11% of its COVID-19 documentation. The WHO recommends implementing transmission mitigation measures like physical distancing and hand hygiene in schools and considering

school closures based on local and/or global evaluations. More specifically, "Deciding to close, partially close or reopen schools should be guided by a risk-based approach to maximize the educational and health benefit for students, teachers, staff, and the wider community, and help prevent a new outbreak of COVID-19 in the community" [57]. A joint publication between the WHO and the Organization for Economic Co-operation and Development (OECD), explains that "The children who suffer most from school closures are reported to be those who live in poor or vulnerable households, notably girls, children with disabilities and marginalized populations" [58]. WHO acknowledges that school closures and mobility restrictions can lead to additional caregiving responsibilities and an increased risk of violence or sexual exploitation, child marriage, and teenage pregnancy for adolescents and young girls. They also recommend that schools "Integrate disease prevention and control in daily activities and lessons. Ensure content is age-, gender-, ethnicity-, and disability-responsive and activities are built into existing subjects" [59].

## Healthcare workforce

More than half of the documents included in the analysis mention the healthcare workforce, but only about 25% of these consider gender. In some cases, this means including the much-touted statistic that 70% of the world's healthcare workers are women. To mitigate the negative impacts on healthcare workers, the WHO recommends that healthcare policymakers ensure appropriate working hours and enforced rest periods, personal protective equipment (PPE) in appropriate sizes for women to limit exposure, and that "women health workers receive equal pay, hold leadership positions and can perform their duties without fear of violence" [60]. It also recommends that health centers designate a gender and equity focused human resources officer to formalize gender dimensions in their decision-making and policy response.

The remaining documents primarily focus on strengthening infection, prevention, and control measures in healthcare settings, increased healthcare worker protections, and ensuring adequate training and PPE for healthcare workers. Other topics include mental health considerations and messages for healthcare workers and managers in health facilities, the inclusion of healthcare workers in research studies, and placing healthcare staff at points-of-entry and other high-risk areas.

The WHO publication on the rights, roles, and responsibilities of healthcare workers during the COVID-19 summarizes the discussion of the considerations and recommendations for this high-risk demographic:

> Health workers should continue to enjoy their right to decent, healthy and safe working conditions in the context of COVID-19. Primary prevention of COVID-19 among health workers should be based on risk assessment and introduction of appropriate measures. Other occupational risks amplified by the COVID-19 pandemic, including violence, harassment, stigma, discrimination, heavy workload and prolonged use of personal protective equipment (PPE) should be addressed. Occupational health services, mental health and psychosocial support, adequate sanitation, hygiene and rest facilities should be provided to all health workers. [61]

## Unpaid care work and labour

The WHO documents acknowledge that women and girls are more likely to be informal caregivers. Outside of the gender considerations, the documents provide recommendations for caregivers of suspected or confirmed COVID-19 patients in the home, including PPE, hand

hygiene, distancing, and airflow guidance. The WHO also recognizes that school closures will increase women's carework and that the increased responsibility may limit their access to services and prevent caregivers from being able to engage in paid work outside their home. The main advice given is that, "The additional care burden associated with COVID-19 needs to be recognized and should be incorporated into policy-making and response measures" [32] and that "planners should emphasize consideration of gender issues, including by supporting women's leadership and recognizing unpaid social care burdens" [48].

Like education, labour is not a major focus of the WHO, but the economic downturn that resulted from the pandemic makes it an important part of the pandemic response. Approximately 14% of the documents analyzed discuss the workforce, business closures, or provide COVID-19 guidance for the workplace, for example: "COVID-19 prevention measures in terms of physical distancing, hand washing, respiratory etiquette and, potentially, thermal monitoring. Teleworking, staggered shifts, and other practices should also be encouraged to reduce crowding" [62]. Among the documents there are also sector-specific labour and workplace guidance for industries including sanitation, ships, aviation, ground crossings, accommodations, food, and people who handle live animals.

As far as gender and labour, the WHO recognizes that women and girls are more likely to work in the informal economy and informal works are more likely to be affected by the adverse labour market effects of the pandemic. One policy guidance document gives the following gender-sensitive advice: "Work with businesses to prepare gender-responsive business continuity plans. [. . .] Ensure sensitivity to single- parent workers and those with additional care responsibilities." [51]

### Risk communication and community engagement (RCCE)

Over 25% of the COVID-19 documents analyzed discuss RCCE. They emphasize educating the public about the pandemic, informing them of public health measures being implemented, and encouraging protective behaviors, such as hand hygiene, respiratory etiquette, social distancing, mask usage (changes over time), and contacting a healthcare provider with questions or concerns or if symptoms develop. Additionally, RCCE is viewed as a tool for combatting the "infodemic" because it "will help limit or stop the spread of rumours about the outbreak and can be used to convey accurate and clear information about COVID-19" [63].

The prevailing RCCE advice given by the WHO is to identify community influencers, leaders, networks, and key individuals that can help with communication and engagement by increasing trust and social cohesion. Women, women's groups, or women's organizations are often included among the suggested influencers. One quote makes this point clear, "The role women play in communities needs to be harnessed in community mobilization efforts" [64].

Some documents not only recommend harnessing the power of women to reach specific audiences, but also acknowledge the importance of messaging that is gender-sensitive and specifically targeted to all population sub-groups [38]. One even discusses how groups of people may be more or less likely to get their information using different platforms, based on their individual characteristics, and recommends choosing, "language and content that matches the platform and speaks to audiences using the platform" [37]. Another document explains that community leaders such as women's groups should be utilized for the diversity of a given community to be represented in the RCCE efforts [65].

### Limitations

We recognize that in this paper we apply a gendered analysis that does not include a meaningful intersectional analysis, and thus we are unable to comment on the other drivers of

inequality which intersect with gender. A related limitation is that our analysis, reflecting the content of WHO's policies, focuses largely on those issues that primarily effect "women and girls" rather than a broader scope to include analysis of the effects of COVID-19 related policies from WHO on men and gender minorities, each of which have had common but differential experiences during the pandemic. A third limitation is that as we undertook a policy analysis of documents from only one year (2020), and did not undertake interviews with policymakers, we are unable to trace policy development over time, or fully elaborate on the dynamics of the organization and context. We were unable to assess the structural, leadership or managerial processes within the international organization to understand whether the permanency of the gender blindness, despite their commitment to gender mainstreaming, is the result of silo-ization of gender within a particular department (such as the gender, equity and rights department) or whether it is systemic, and what barriers remain to more meaningfully integrating gender mainstreaming across WHO policies. Thus, our analysis is a reflection of a narrow series of WHO's policy outputs during the first year of the COVID-19 pandemic rather than its processes or interactions with the broader context.

## Discussion

There are many instances in which the WHO incorporates gender and gendered impact areas throughout its COVID-19 policy documentation. However, whether it meets its own gender mainstreaming standard of "assessing the implications for women and men of any planned action, including legislation, policies or programs, in all areas and at all levels" [66] is debatable. While it is perhaps unrealistic to expect all documents to include a gender focus, it is alarming that less than one fifth of documentation related to COVID-19 explicitly mentions gender. This is in spite of the fact that WHO has committed in its gender strategy to ensuring that all programmes analyze the role of sex and gender in their work and develop appropriate gender specific responses [67]. Of course including gender considerations in policy documents is but one way to implement mainstreaming, further actions require greater investment, exemplary action by leadership, and challenging the "deep structure of gender bias" within the institutional culture [4, 68].

One explanation for limited application may be that the WHO's approach to gender mainstreaming, with its limited definition and focus on technical process as opposed to power relations, is not fit for purpose–as feminist critiques of gender mainstreaming in global health have previously suggested [27]. Structural constraints may also include: that the tyranny of the urgent during the first phases of the pandemic meant that gender was, as in past instances, left at the margins [69]; (willful) historic amnesia to gendered considerations within the organization, partly due to lack of gender representative decision-making, resulted in a lack of awareness by those crafting health emergency related policy to the gendered effects of all health activities [19]; the pathology of the organization which fails to consider women both formally in its policies and informally in its modus operandi [4]; a narrow interpretation of the WHO's mandate to be on the "highest attainable standard of health" with a predominant focus on biomedical and technical interventions, rather than addressing the social determinants of health; inter-UN mission separation, with WHO assuming gender to be the sole mandate of UN Women; an active decision by those crafting policy to try and create content which is easy for all member states to agree to and work towards, cognizant that other areas of sex and gender related policy have been contentious amongst some member states, for example with regards to abortion [70]. It is likely that a combination of these factors have structured the limited application of gender mainstreaming within WHO's COVID-19 response.

The slightly higher consideration of sex in WHO documents reflects a continued focus on biological (sex), as opposed to social or economic (gender) drivers of health inequities. The areas with the highest proportion of documents—SRH, pregnancy and maternal healthcare—reflect a narrow, technical application of gender mainstreaming common in global health, which reduces women to their reproductive function while ignoring the health needs of those women who are not of reproductive age, as well as of sexual minorities. While the focus on biological drivers of health inequities and specific health priorities can partly be explained by WHO's positioning and remit as a health focused organization, COVID-19 has made explicit the relationships between social and economic and health inequities, demanding greater attention be paid to these intersections alongside critical analysis of the power inequities that structure them.

WHO documentation on COVID-19 does include passing mention of the influence of gendered power relations, roles and norms. Documents analyzed here also consider, in an albeit limited way, COVID-19's effects on migration, education and unpaid care work. As such, WHO documentation touches on two of the three (gender-based violence and unpaid care, but not economic insecurity) priorities highlighted in UN Women's definition of gender sensitive measures noted above. This suggests that COVID-19 may present an opportunity for WHO, and global health more generally, to broaden its approach to gender mainstreaming to be both more holistic and transformative. Previous analysis has documented how gender mainstreaming has been advanced within UN agencies through strategic entry points–COVID-19 could potential provide such a point of convergence around gender [27]. This would require recognition of the extent to which these gaps remain for such omissions to be counteracted, and should form a critical element of WHO reform.

Given that WHO's internal review of its gender mainstreaming activity demonstrated the limited impact of efforts to implement the WHO gender strategy, combined with the surge in awareness of the gendered effects of COVID-19, could have provided a perfect opportunity for more meaningful engagement with gender mainstreaming across all policies [71]. Importantly, we have seen strides to try and further gender mainstream the Health Emergencies Programme (HEP) of the WHO during the pandemic, such as through the establishment of a gender working group within the division, and commitment to gender engagement through civil society groups through the Director General's office, although most of this work has occurred subsequent to the data collection for this paper.

Healthcare workers and the healthcare workforce is the most discussed of gendered impact areas overall, which is in line with the mandate of the WHO, and with recent efforts by the organization to focus on workforce issues, such as the publication of a gender analysis of the global health workforce and the designation of 2020 as the year of the nurse and midwife [72]. However, only about 25% of these documents talk about women and gender, despite the fact that 70% of healthcare workers globally are women (a statistic frequently repeated by the WHO). This is one area in which the WHO can do more to discuss the distinct implications of an issue for women, men, and gender minorities, and to tie the impact of COVID-19 on the healthcare workforce into their ongoing workstreams on gender and workforce. However, this is problematic to implement given the siloed work amongst programmes within the WHO, limited to activity often based on donor funding for such efforts. As such, whilst the workforce programme of WHO may have external funding for gender mainstreaming, this was not the case for HEP at the start of the pandemic. Thus, fragmented funding within the WHO remains a key problem for meaningful organization-wide gender mainstreaming [4].

A further area which is problematic when considering gender mainstreaming within the WHO's response is RCCE. Though over 40% of the documents that discuss RCCE do discuss gender, most of this effect comes from recommendations that women and their networks be

utilized to their full effect to support the COVID-19 response: policymakers should take advantage of women's organizations, to serve as representatives and influencers to their communities for improved community engagement. While representation and diversity in RCCE is important, it is concerning that women are being asked to assist in risk communication and community engagement without being compensated for their time or labour and without having their specific needs considered or prioritized in the overall pandemic response. The 'problem' is perceived as poor risk communication, with women's unpaid labour then positioned as the solution. As Bacchi points out, how problems are represented reflects political influences and power configurations [30]. In this case, while the WHO recognizes the informal work that women do during pandemics, it takes limited efforts to mitigate, reduce or redistribute such labour to place less burden on women's shoulders, positioning it as a solution as opposed to problems (or solution as opposed to threat), and therefore seeks to place further unpaid work on their shoulders [18]. This mirrors the responsibilisation of women to implement global health policy as witnessed during other epidemics [19], and the incremental recognition that gender engagement can provide a pertinent solution to filling the labour gap in public health [18].

There are many instances in which the WHO asks national policymakers to ensure that a given policy is gender-responsive or to consider gendered impacts when formulating policy, but these serve more as reminders than concrete recommendations. While there is no one-size-fits-all solution to these complex social and cultural issues, providing suggestions for specific policies and actions that can be taken to address and prevent these issues would be a more effective way to ensure a gender-sensitive pandemic response.

We must take the consideration of gender within WHO in the context of its waning role in global health governance, with many governments turning their backs on the institution during the pandemic, charting their own courses domestically and failing to heed to the solidarity, cooperative stance of the institution. In this light, pushing for greater gender transformation within the institution may have limited effect on outcomes domestically, and efforts may be better placed elsewhere. However, as the normative authority on global public health, it is incumbent upon the WHO to make every effort to mainstream gender and set precedent for how best to gender mainstream pandemic response policy [4]. Throughout the publications, there is an acknowledgment of the disproportionate impacts that COVID-19 is having and will have on women and girls, but acknowledging these issues is only the first step. The WHO might seize the opportunity created by COVID-19, and its starkly gendered effects, to not only reaffirm its commitment to gender mainstreaming, but shift to a more transformative approach that includes acting on, as well as assessing, gender inequities.

## Conclusion

As a key determinant of health, gender should be prioritized in the global response to COVID-19. Lessons from past pandemics show that when gender is not considered, the effects on women's health, safety, and livelihoods can be devastating and persistent. By May 2020, the WHO was explicitly emphasizing the importance of a gender-responsive and non-discriminatory policy response to COVID-19. The Director-General asked that Member States collect sex-disaggregated data, prevent and respond to domestic violence, encourage access to SRH services, protect healthcare workers, and ensure equitable access to COVID-19 testing and treatment. In this analysis, we find that over half of the WHO COVID-19 publications do not discuss women or gender at all and those that do tend to talk about women in terms of their biological sex and not in terms of socially-constructed gender differences. These social and behavioral differences can lead to adverse effects from the outbreak and the pandemic

mitigation measures. Going forward, the WHO should increase its gender mainstreaming efforts and incorporate gender considerations more often and in more policy areas in order to establish gender as a priority in pandemic response.

## Supporting information

**S1 Appendix. List of all WHO COVID-19 documents included and analyzed.**
(DOCX)

## Author Contributions

**Conceptualization:** Julia Smith, Clare Wenham.

**Data curation:** Emily Tomsick.

**Formal analysis:** Emily Tomsick, Clare Wenham.

**Funding acquisition:** Julia Smith, Clare Wenham.

**Investigation:** Julia Smith.

**Methodology:** Julia Smith, Clare Wenham.

**Project administration:** Julia Smith.

**Supervision:** Julia Smith, Clare Wenham.

**Validation:** Julia Smith.

**Visualization:** Emily Tomsick.

**Writing – original draft:** Emily Tomsick.

**Writing – review & editing:** Julia Smith, Clare Wenham.

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
