## [Decision Letter · Decision Letter 0]

4 Jan 2022

PGPH-D-21-00460

A gender analysis of the World Health Organization’s COVID-19 guidance and policies

Dear Dr. Smith,

Thank you for submitting your interesting manuscript to PLOS Global Public Health. We sincerely apologize for the delay in sharing a decision on this paper; unfortunately we were unable to secure a second reviewer for quite some time, and so I have provided a review as academic editor for this piece. The decision is for a major revision, and we invite you to submit a revised version of the manuscript that addresses the points raised during the review process. We will be sure to turn around a decision after resubmission quickly. Thank you for your patience and if you decide to proceed, we look forward to receiving the resubmission. 

EDITOR:

This is an important policy analysis on considerations of sex and gender in WHO policy documents. One of the issues that I observed while reviewing this paper was how to best understand the ways in which actors should utilize terms such as “women and girls”. My understanding from reading this piece is that solely focusing on “women and girls” deemphasizes the focus on sexual minorities and gender minorities (p. 9, where the authors call the primary use of this term problematic); however, the authors use women and girls periodically throughout the manuscript (for example, p. 23 or women and gender noted on p. 21) to discuss the lack of gender-mainstreaming in WHO’s policy documents. It would be important to clarify the authors’ stance so that readers are clear on where the gaps lie.

Please provide additional details on the content analysis – please see Reviewer 1’s suggestions on this.

In the Results section, there appeared to be inconsistent use of citations for documents – please review carefully to make sure that documents are cited wherever feasible. Some of the quotes can be reduced in length

R1 makes an important point about providing some additional depth in the analysis in terms of what these findings mean for issues of power in policy and decision-making processes within global health organizations; I recognize that this is not the primary objective of the paper, but the authors could provide some limited additional analysis in this direction in order to strengthen the discussion.

We look forward to receiving your revised manuscript.

Kind regards,

Veena Sriram

Academic Editor

Journal Requirements:

1. Please ensure your Methods are described in sufficient detail for another researcher to reproduce the experiments described. Specifically, please ensure you have provided a list of policy documents included in your analysis in the supporting information.

2. Please provide separate figure files in .tif or .eps format only.  Please ensure that all files are under our size limit of 20MB.  

For more information about how to convert your figure files please see our guidelines: Once you've converted your files to .tif or .eps, please also make sure that your figures meet our format requirements

3. Please update the completed 'Competing Interests' statement, including any COIs declared by your co-authors. If you have no competing interests to declare, please state "The authors have declared that no competing interests exist".

4. Please amend your detailed Financial Disclosure statement. This is published with the article, therefore should be completed in full sentences and contain the exact wording you wish to be published.

ii). State the initials, alongside each funding source, of each author to receive each grant.

iii). State what role the funders took in the study. If the funders had no role in your study, please state: “The funders had no role in study design, data collection and analysis, decision to publish, or preparation of the manuscript.”

5. Please provide the link of WHO website in your Data Availability in the system.

Additional Editor Comments (if provided):

Please accept our apologies for the delay in sharing a decision on this paper. We were unable to secure a second reviewer for quite some time, and so I decided to provide the review myself. Please see both the attached documents for detailed reviews of the paper.

Reviewers' comments:

Reviewer's Responses to Questions

**Comments to the Author**

1. Does this manuscript meet PLOS Global Public Health’s publication criteria? Is the manuscript technically sound, and do the data support the conclusions? The manuscript must describe methodologically and ethically rigorous research with conclusions that are appropriately drawn based on the data presented.

Reviewer #1: Yes

2. Has the statistical analysis been performed appropriately and rigorously?

Reviewer #1: Yes

3. Have the authors made all data underlying the findings in their manuscript fully available (please refer to the Data Availability Statement at the start of the manuscript PDF file)?

Reviewer #1: Yes

4. Is the manuscript presented in an intelligible fashion and written in standard English?

Reviewer #1: Yes

5. Review Comments to the Author

Reviewer #1: The reviewer has attached a file with brief and specific comments and questions with regards to the Introduction, Methods and Results and these are intended to enhance and strengthen the analysis presented in paper.

6. PLOS authors have the option to publish the peer review history of their article (what does this mean?). If published, this will include your full peer review and any attached files.

**Do you want your identity to be public for this peer review?** For information about this choice, including consent withdrawal, please see our Privacy Policy.

Reviewer #1: No

---

## [Decision Letter · Decision Letter 1]

29 Mar 2022

PGPH-D-21-00460R1

A gender analysis of the World Health Organization’s COVID-19 guidance and policies

Dear Dr. Smith,

Thank you for submitting your manuscript to PLOS Global Public Health. Upon considering the feedback from the reviewer, please find some suggested minor changes below before we can accept. We invite you to submit a revised version of the manuscript that addresses the points raised during the review process.

We look forward to receiving your revised manuscript.

Kind regards,

Veena Sriram

Academic Editor

Journal Requirements:

1. Please change "Figure one" to "Fig 1" where you refer your figure on page 9.

Additional Editor Comments (if provided):

- R1 notes that the authors should have a deeper discussion of social and political processes - my sense is that the authors have addressed this concern within the boundaries of the paper. I would suggest reviewing the reviewer’s feedback on this point and suggested papers to see if there are any other places within the manuscript to strengthen this particular concern.

- Page 6 – should be Microsoft Excel instead of excel?

- Page 23 – The point about right wing populism in 2020 – my sense is that there is a longer trajectory of that trend that predates 2020, and so would suggest reframing.

- Please consider R1’s suggestion about citing Ravindran et al 2021.

Reviewers' comments:

Reviewer's Responses to Questions

**Comments to the Author**

1. If the authors have adequately addressed your comments raised in a previous round of review and you feel that this manuscript is now acceptable for publication, you may indicate that here to bypass the “Comments to the Author” section, enter your conflict of interest statement in the “Confidential to Editor” section, and submit your "Accept" recommendation.

Reviewer #1: (No Response)

2. Does this manuscript meet PLOS Global Public Health’s publication criteria? Is the manuscript technically sound, and do the data support the conclusions? The manuscript must describe methodologically and ethically rigorous research with conclusions that are appropriately drawn based on the data presented.

Reviewer #1: Yes

3. Has the statistical analysis been performed appropriately and rigorously?

Reviewer #1: (No Response)

4. Have the authors made all data underlying the findings in their manuscript fully available (please refer to the Data Availability Statement at the start of the manuscript PDF file)?

Reviewer #1: Yes

5. Is the manuscript presented in an intelligible fashion and written in standard English?

Reviewer #1: Yes

6. Review Comments to the Author

Reviewer #1: See attached file for further review comments

7. PLOS authors have the option to publish the peer review history of their article (what does this mean?). If published, this will include your full peer review and any attached files.

**Do you want your identity to be public for this peer review?** For information about this choice, including consent withdrawal, please see our Privacy Policy.

Reviewer #1: No

---

## [Editor Report · Decision Letter 2]

27 May 2022

A gender analysis of the World Health Organization’s COVID-19 guidance and policies

PGPH-D-21-00460R2

Dear Dr Smith,

We are pleased to inform you that your manuscript 'A gender analysis of the World Health Organization’s COVID-19 guidance and policies' has been provisionally accepted for publication in PLOS Global Public Health. Many thanks for your patience throughout this process! 

Best regards,

Veena Sriram

Academic Editor